# Interplay between the Human Microbiome and Biliary Tract Cancer: Implications for Pathogenesis and Therapy

**DOI:** 10.3390/microorganisms11102598

**Published:** 2023-10-20

**Authors:** Cheng Ye, Chunlu Dong, Yanyan Lin, Huaqing Shi, Wence Zhou

**Affiliations:** 1The First Clinical Medical College, Lanzhou University, Lanzhou 730000, China; yech17@163.com (C.Y.); dongcl1122@126.com (C.D.); ldyy_linyy@lzu.edu.cn (Y.L.); lzushq@163.com (H.S.); 2Department of General Surgery, The First Hospital of Lanzhou University, Lanzhou 730000, China; 3Department of General Surgery, The Second Hospital of Lanzhou University, Lanzhou 730000, China

**Keywords:** biliary tract cancer, cholangiocarcinoma, microbiome, metabolite

## Abstract

Biliary tract cancer, encompassing intrahepatic and extrahepatic cholangiocarcinoma as well as gallbladder carcinoma, stands as a prevalent malignancy characterized by escalating incidence rates and unfavorable prognoses. The onset of cholangiocarcinoma involves a multitude of risk factors and could potentially be influenced by microbial exposure. The human microbiome, encompassing the entirety of human microbial genetic information, assumes a pivotal role in regulating key aspects such as host digestion, absorption, immune responses, and metabolism. The widespread application of next-generation sequencing technology has notably propelled investigations into the intricate relationship between the microbiome and diseases. An accumulating body of evidence strongly suggests a profound interconnection between biliary tract cancer and the human microbiome. This article critically appraises the existing evidence pertaining to the microbiome milieu within patients afflicted by biliary tract cancer. Furthermore, it delves into potential mechanisms through which dysregulation of the human microbiome could contribute to the advancement of biliary tract cancer. Additionally, the article expounds on its role in the context of chemotherapy and immunotherapy for biliary tract cancer.

## 1. Introduction

Biliary tract cancer refers to a spectrum of invasive adenocarcinomas, including cholangiocarcinoma and gallbladder carcinoma [1]. Cholangiocarcinoma is further categorized into intrahepatic cholangiocarcinoma (ICC) and extrahepatic cholangiocarcinoma (the latter further divides into perihilar and distal cholangiocarcinoma). The incidence of biliary tract cancer exhibits variation across subgroups and geographic regions, exhibiting a notable rise over the years, particularly for ICC. In the United States, ICC’s incidence escalated from 0.44 to 1.18 cases per 100,000, while extrahepatic cholangiocarcinoma experienced a more modest increase from 0.95 to 10.2 per 100,000 over a 40-year period [2]. Cholangiocarcinoma has a poor prognosis and is usually identified at advanced stages. This typically occurs when the primary tumor reaches a substantial size, resulting in a sizable liver mass, or when jaundice develops due to obstruction in the biliary tree [3].

Epidemiological investigations have elucidated the involvement of multiple risk factors in the genesis of cholangiocarcinoma. Bile duct cysts, primary sclerosing cholangitis, hepatolithiasis, cholelithiasis, and choledocholithiasis all exhibit associations with cholangiocarcinoma [4]. In addition, the liver fluke *Opisthorchis viverrini* is the main cause of cholangiocarcinoma in Southeast Asia [5]. Host genetics, lifestyle choices, environmental exposures, and other factors also influence the course of cholangiocarcinoma [4]. The term “microbiome” pertains to the cumulative genetic makeup of microorganisms within a specific environment, carrying a pivotal role in immune regulation and safeguarding the host against pathogenic microbes [6]. Disruption of the gut microbiome has been implicated in an array of conditions, encompassing cancer and metabolic disorders [7,8].

The advent of next-generation sequencing (NGS) technology has made microbiome analysis more convenient, thereby fostering a substantial upsurge in research investigating the interplay between the human microbiome and cancer. Mounting evidence underscores the potential impact of the microbial milieu on individuals with biliary tract cancer [9]. Notably, individuals afflicted with biliary tract cancer often exhibit elevated levels of *Enterobacteriaceae* but diminished levels of *Clostridia*, including *Faecalibacterium* and *Coprococcus*. *Enterobacteriaceae* see enrichment within the fecal samples of those with biliary tract cancer, with over half of the *Enterobacteriaceae* identified in bile matching those present in fecal samples at the operational taxonomic unit (OTU) level. These findings collectively hint at the potential contribution of fecal microbiota dysbiosis to the development of biliary tract cancer [10]. Within this review, we delve into the intricate relationship between the digestive tract microbiome and biliary tract tumors, dissecting the role and significance of the digestive tract microbiome in the realm of biliary tract cancer treatment.

## 2. Biliary Tract Cancer and the Human Microbiome

### 2.1. The Microbiome and Intrahepatic Cholangiocarcinoma

Intrahepatic cholangiocarcinoma (ICC) stands as a highly malignant form of primary liver cancer, originating from the epithelial cells of intrahepatic bile ducts [1,11]. The scarcity of typical clinical symptoms leads to a mere 22% of patients qualifying for surgical intervention [12]. The emergence of cholangiocarcinoma is intricately tied to the carcinogenic influence of chronic biliary inflammation [13]. While primary sclerosing cholangitis, Caroli’s disease, and choledochal cysts share associations with all cholangiocarcinoma variants, cirrhosis, non-alcoholic fatty liver disease (NAFLD), and hepatitis B exhibit a stronger link with ICC [14].

Studies have unveiled a connection between the occurrence of ICC and gut dysbiosis [15,16,17]. Jia et al. analyzed gut microbiota and bile acid metabolism in patients with ICC, systematically demonstrating the relationship among gut microbiota, bile acid, and cytokine profiles. Comparatively, ICC displayed the highest α-diversity and β-diversity when juxtaposed against patients with hepatocellular carcinoma, liver cirrhosis, and healthy individuals. Notably enriched in ICC were *Actinomyces*, *Lactobacillus*, *Peptostreptococcaceae*, and *Alloscardovia*. Furthermore, the glycoursodeoxycholic acid and tauroursodeoxycholic acid (TUDCA) plasma-stool ratios were significantly increased in ICC, with the genera *Lactobacillus* and *Alloscardovia* exhibiting positive correlations with the TUDCA plasma-stool ratio. These biomarkers could be used to differentiate ICC from hepatocellular carcinoma (HCC) [18]. Deng et al. conducted a comprehensive analysis encompassing a cohort of 40 healthy volunteers, 143 HCC patients, and 46 cholangiocarcinoma patients based on fecal 16S rRNA sequencing. The cholangiocarcinoma group displayed increased gram-negative bacteria levels and inflammatory markers compared to the HCC group. They established the gut microbiome-based model for liver cancer prediction and screening, divulging a correlation between primary liver cancer-related gut microbiome characteristics and unfavorable inflammatory response markers [19]. Similarly, Zhang et al. explored a gut microbiota model covering the genera *Burkholderia-Caballeronia-Paraburkholderia, Faecalibacterium*, and *Ruminococcus_1* (B-F-R) for early cholangiocarcinoma diagnosis [20]. Moreover, oral microbiota-targeted biomarkers have emerged as effective noninvasive diagnostic tools for cholangiocarcinoma [21]. In instances of mice afflicted with primary sclerosing cholangitis (PSC) and colitis, compromised gut barrier function facilitated the infiltration of gut-derived bacteria and lipopolysaccharides (LPS) into the liver. The intestinal microbiome spurred CXCL1 expression in hepatocytes via TLR4-dependent mechanisms, fostering the accumulation of CXCR2+ polymorphonuclear myeloid-derived suppressor cells (PMN-MDSC). This mechanism created an immunosuppressive environment in hepatocytes, thereby promoting ICC development [22]. Thus, the gut microbiome holds promise as a potential ICC biomarker.

Beyond the digestive tract microbiome, other facets of the microflora within ICC patients have garnered attention. Chai et al. performed 16S rRNA sequencing, single-cell RNA sequencing (scRNA-seq), and multilayer validation on cholangiocarcinoma tissues. They verified the presence of microbial DNA in tissues via staining, fluorescence in situ hybridization (FISH), and transmission electron microscopy (TEM). Intratumoral bacteria manifest across multiple cell types, as evidenced by scRNA-seq [23]. Chng et al. found that a distinct and tissue-specific microbiome dominated by the families *Dietziaceae*, *Pseudomonadaceae*, and *Oxalobacteraceae* was observed in the bile duct tissues. Compared to paracancerous tissue and normal liver tissue, substantial variations in colonized flora were noted in bile duct carcinoma tissue, with *Stenotrophomonas* species showing a significant increase. The enrichment of specific enteric bacteria (*Bifidobacteriaceae*, *Enterobacteriaceae*, and *Enterococcaceae*) correlated with parasite-associated cholangiocarcinoma [24]. Moreover, intratumoral microbial composition held relevance in chemotherapy resistance within cholangiocarcinoma [25]. Lee et al. isolated bacterial-derived extracellular vesicles from the plasma of biliary tract cancer patients, dissecting microbiome composition via 16S rDNA metagenomic analysis. Microflora composition showed variable percentages from phylum to genus level. They formulated a predictive model for biliary tract cancer based on variations in blood microbial composition. However, blood microbiome exploration remains relatively nascent, necessitating further research to comprehend alterations and underlying mechanisms [26].

### 2.2. The Microbiome and Extrahepatic Cholangiocarcinoma

Choledocholithiasis exhibits a clear correlation with extrahepatic cholangiocarcinoma and serves as a potential risk factor for its development. This association may be attributed to biliary bacterial infections [14]. Jan Bednarsch et al. conducted a study involving intraoperative bile samples from patients with hilar cholangiocarcinoma to perform microbial cultures. The results indicated a substantial colonization of the bile ducts by bacteria. Among the most prevalent bacteria identified in the bile ducts were *Enterococcus faecalis* (38.8%, 31/80), *Enterococcus faecium* (32.5%, 26/80), *Enterobacter cloacae* (16.3%, 13/80), and *Escherichia coli* (11.3%, 9/80). Notably, reduced susceptibility of these bacteria to intraoperative antibiotic prophylaxis was identified as an independent predictor of postoperative abdominal infections [27]. Di Carlo et al. demonstrated that an unprecedented increase in *E. coli* within the bile of cholangiocarcinoma patients corresponded to decreased survival rates. This suggests that certain strains isolated from bile samples may be considered as contributing to the group of risk factors in the carcinogenesis and/or progression of hepatobiliary malignancies [28]. In addition, isolated biliary candidiasis may be associated with a poor prognosis in patients with unresectable cholangiocarcinoma [29].

The utilization of next-generation sequencing technology has greatly facilitated the exploration of the intestinal and biliary microflora. Extensive investigations have been conducted on the bile microbiome of individuals with extrahepatic cholangiocarcinoma [30,31,32]. These studies involve the collection of bile samples through ERCP from patients with cholangiocarcinoma and cholangiolithiasis for 16S rRNA sequencing analysis. The analysis outcomes revealed that the most abundant genera within the biliary microflora were *Enterococcus*, *Streptococcus*, *Bacteroides*, *Klebsiella*, and *Pyramidobacter*. In comparison to cholangiolithiasis cases, levels of *Bacteroides*, *Geobacillus*, *Meiothermus*, and *Anoxybacillus* genera were significantly elevated in the biliary microbiota of patients with extrahepatic cholangiocarcinoma [30,31]. Noteworthy discrepancies exist in the microbial communities present in the bile of choledocholithiasis and cholangiocarcinoma patients. These bacteria potentially play a partial role in the onset of cholangiocarcinoma and could serve as novel biomarkers for this condition [33].

Aviles-Jimenez et al. collected bile duct epithelial cells via brushing from 100 patients with extrahepatic cholangiocarcinoma and 100 patients with benign biliary diseases during ERCP. Their analysis of DNA extractions revealed reduced levels of *Nesterenkonia* but increased levels of *Methylophilaceae*, *Fusobacterium*, *Prevotella*, *Actinomyces*, *Novosphingobium*, and *H. pylori* in extrahepatic cholangiocarcinoma cases. They verified the potential role of *H. pylori* in the development of extrahepatic cholangiocarcinoma [34]. Miyabe et al. identified a distinctive microbial signature in the bile of patients with prolonged PSC duration or those with cholangiocarcinoma, suggesting a role for microbiota-driven inflammation in the pathogenesis or progression of perihilar cholangiocarcinoma [35]. Thus, the microbiome of the digestive tract emerges as a pivotal factor in the development of extrahepatic cholangiocarcinoma. Table 1 presents studies evaluating microbial composition in patients with biliary tract cancer.

### 2.3. The Microbiome and Gallbladder Cancer

Gallbladder cancer is a prevalent malignancy affecting the biliary tract, and its prognosis is notably grim when diagnosed at an advanced stage due to its aggressive nature and limited treatment avenues. Prolonged chronic inflammation plays a pivotal role in the development of gallbladder cancer, irrespective of whether it originates from gallstones or other sources [36]. Multiple studies have indicated a heightened risk of gallbladder cancer in the presence of bacterial infections. Notably, analysis of bile samples from gallbladder cancer patients revealed a substantial increase in bacterial taxa [37]. It is plausible that chronic bacterial infection of the bile, leading to the production of carcinogenic precursors, is among the causative factors underlying the emergence of gallbladder carcinoma [38]. The persistent presence of certain bacteria triggers chronic inflammation, giving rise to toxins and metabolites with carcinogenic potential. These elements contribute to the transformation of gallbladder epithelial cells [39].

Tsuchiya et al. conducted a study comparing bacteria found in bile samples from gallbladder cancer patients and those with cholelithiasis. The study highlighted that the incidence of bacterial infection in bile was 42.9 percent for gallbladder cancer patients, compared to 13.3 percent for cholelithiasis patients. The dominant species identified in the bile of gallbladder cancer patients included *Fusobacterium nucleatum*, *Escherichia coli*, and *Enterobacter* sp., while the bile from cholelithiasis patients primarily contained *Escherichia coli*, *Salmonella* sp., and *Enterococcus gallinarum* [40]. Another study hints at a potential correlation between a dysbiotic bile microbiome and the development of chronic calculous cholecystitis and gallbladder cancer. Patients with chronic cholecystitis and an imbalanced microbiome pattern exhibited larger gallstones and notable epithelial abnormalities, considered precancerous conditions. These findings suggest the potential involvement of *Enterobacteriaceae*, including *Klebsiella*, in gallbladder carcinogenesis [41].

The chronic presence of *Salmonella typhi* in gallbladder disease may contribute to the onset of gallbladder cancer [42]. Meta-analysis indicated that chronic *Salmonella typhi* infection correlated with an increased risk of gallbladder cancer, representing a significant risk factor for the condition [43,44]. Experimental evidence underscores the ability of *Salmonella enterica* to facilitate the transformation of genetically predisposed cells, ultimately acting as a causative agent of gallbladder cancer. This bacterium induces malignant transformation in susceptible mice, murine gallbladder organoids, and fibroblasts by triggering TP53 mutations and c-MYC amplification. Mechanistically, the activation of MAPK and AKT pathways, mediated by *Salmonella enterica* effectors released during infection, is instrumental in both initiating and sustaining transformation [45]. Beyond *Bacillus typhoid*, infection with specific strains of *H. pylori* has also been linked to an increased risk of biliary tract cancer [46,47]. Experimental data suggests that *Helicobacter bilis* infection activates transcription factors such as NFKB, leading to enhanced angiogenesis through VEGF production. The involvement of *Helicobacter bilis* infection may be significant in biliary tract malignancies [48].

In a study by Song et al., mucosal DNA extraction and metagenomic sequencing were employed to compare the microbiota between patients with chronic calculous cholecystitis and gallbladder cancer. This analysis revealed substantial differences in biliary microbial composition and gene function between the two groups. *Peptostreptococcus stomatis* and *Enterococcus faecium* were identified as potential contributors to the progression of gallbladder cancer [49]. Furthermore, a notable connection was established between the relative abundance of specific microbes and the overall survival prognosis of patients with pancreaticobiliary tract cancer [50]. Hence, targeting bacterial infections through anti-inflammatory treatments and hygiene practices could potentially mitigate the incidence of gallbladder cancer.

**Table 1 microorganisms-11-02598-t001:** Studies evaluating microbial composition in patients with biliary tract cancer.

Author, Year	Biological Specimens	Sampling Methods	Tumor Site and Size	Main Conclusion
Chen, 2019 [30]	bile	ERCP	dCCA, 8	*Proteobacteria*, *Firmicutes*, *Bacteroidetes*, and *Actinobacteria* are the most dominant phyla in the bile. *Gemmatimonadetes*, *Nitrospirae*, *Chloroflexi*, *Latescibacteria*, and *Planctomycetes* in the phylum increase in dCCA patients compared with the onset of common bile duct stones patients.
Avilés-Jiménez, 2016 [34]	biliary duct epithelial cells	Brushing during ERCP	eCCA, 100	Phylum *Proteobacteria* dominated all samples (60.4% average). *Nesterenkonia* decreased, whereas *Methylophilaceae*, *Fusobacterium*, *Prevotella*, *Actinomyces*, *Novosphingobium*, and *H. pylori* increased in eCCA. Predicted associated functions showed an increased abundance of *H. pylori* virulence genes in eCCA.
Saab, 2021 [31]	bile	ERCP	eCCA, 28	*Proteobacteria* did not significantly differ between eCCA patients and controls. The most abundant genera were *Enterococcus*, *Streptococcus*, *Bacteroides*, *Klebsiella*, and *Pyramidobacter* in eCCA’s biliary microbiota. Levels of *Bacteroides*, *Geobacillus*, *Meiothermus*, and *Anoxybacillus* genera were significantly higher in eCCA patients’ biliary microbiota, without an associated disease, in comparison with controls.
Li, 2022 [32]	bile	ERCP	pCCA, 14 dCCA, 9	The top three biomarkers for pCCA at the genus level were *Pseudomonas*, *Sphingomonas*, and *Halomonas*; for dCCA, they were *Streptococcus*, *Prevotella*, and *Halomonas*.
Miyabe, 2022 [35]	Bile and stool	ERCP	CCA (mainly pCCA), 49	Increased species richness and abundance of *Fusobacteria* were correlated with the duration of PSC and characterized the biliary microbiota in CCA.
Ito, 2022 [10]	Bile and stool	ERCP	iCCA, 12 eCCA, 12 GBC, 6	A higher *Enterobacteriaceae* abundance and a lower *Clostridia* abundance, including that of *Faecalibacterium* and *Coprococcus*, in the BTC patients than in the other subjects. A bile-isolated strain possessed the carcinogenic bacterial colipolyketide synthase-encoding gene.
Di Carlo, 2019 [28]	bile	ERCP	CCA, 42 GBC, 5	*E. coli* and *P. aeruginosa* were significant negative predictors of CCA. About GBC, there were no significant correlations with *E. coli*, *K. pneumoniae*, or *P. aeruginosa*.
Pomyen, 2023 [51]	stool	-	iCCA, 19	Two *Veillonella* species were found to be more abundant in iCCA samples and could distinguish iCCA from HCC and healthy controls. *Ruminococcus gnavus* was depleted in iCCA patients and could distinguish HCC from iCCA samples. High *Veillonella* genus counts in the iCCA group were associated with enriched amino acid biosynthesis and glycolysis pathways.
Chai, 2023 [23]	tissues	surgery	iCCA, 99	The most abundant bacterial orders include *Burkholderiales*, *Pseudomonadales*, *Xanthomonadales*, *Bacillales*, and *Clostridiales*. The content of *Paraburkholderia* fungorum was significantly higher in the paracancerous tissues.
Deng, 2022 [19]	fecal	-	CCA, 46	*Gammaproteobacteria* were significantly higher in both gemcitabine- and cisplatin-resistance groups compared to sensitive groups.
Jia, 2020 [18]	stool and blood	-	iCCA, 28	The abundances of four genera (*Lactobacillus, Actinomyces, Peptostreptococcaceae*, and *Alloscardovia*) were increased in patients with ICC compared with those in patients with hepatocellular carcinoma or liver cirrhosis and in healthy individuals. The glycoursodeoxycholic acid and tauroursodeoxycholic acid (TUDCA) plasma-stool ratios were obviously increased in patients with ICC.
Chng, 2016 [24]	tissue	-	CCA, 60	A distinct, tissue-specific microbiome dominated by the bacterial families *Dietziaceae*, *Pseudomonadaceae*, and *Oxalobacteraceae* was observed in bile duct tissues. Several bacterial families, with a significant increase in *Stenotrophomonas* species distinguishing tumors from paired normals.

## 3. The Effect of Dysbiosis on Biliary Tract Cancer and Its Precancerous Lesions

The impairment of intestinal barrier function facilitates the buildup of gut-derived bacteria and LPS within the portal vein. This process triggers the accumulation of myeloid-derived suppressor cells (MDSC) via TLR4-dependent mechanisms, thus fostering immune evasion and driving the advancement of CCA. Fecal microbiota transplantation (FMT) has been shown to encourage MDSC accumulation in the liver using fecal samples from mice with intestinal disorders. However, pre-treating donor mice with neomycin to eliminate gram-negative bacteria counters this effect. The MDSC induced by gram-negative bacteria assumes a pivotal role in amplifying cholangiocarcinoma progression [22,52]. 

Primary sclerosing cholangitis (PSC), an immune-associated cholangitis, is linked to a heightened risk of cholangiocarcinoma and gallbladder cancer [53,54]. Evidence indicates that the dysregulation of the microflora is implicated in the pathogenesis of PSC [55,56]. Patients with PSC display disruptions in the upper digestive tract and bile duct microbiota. Biliary dysbiosis is correlated with elevated levels of the proinflammatory and potentially cancerogenic agent taurolithocholic acid [57].

Over the past decade, an array of studies involving both humans and animal models have underscored the role of the microbiome in various segments of the gastrointestinal tract in the development of gallstone disease [58]. Changes in the gastrointestinal microbiome may reshape the pathogenesis of cholesterol gallstone formation. Alterations in the oral microbiome influence the expression of mucin genes via immune modulation. This, in turn, modifies the accumulation of mucin gel, thereby heightening the risk of bile supersaturation and ultimately accelerating the process of gallstone formation [59,60]. The presence of *Helicobacter pylori* infection contributes to the formation of cholecystic polyps and gallstones [61] and affects the pathophysiology of gallstone formation along with its associated complications such as cholecystitis, cholangitis, pancreatitis, and biliary cancer [62]. Intestinal bacteria (*Clostridium*, *Bifidobacterium*, *Peptostreptococcus*, *Bacteroides*, *Eubacterium*, and *Escherichia coli*) involved in bile acid oxidation and epimerization can disrupt enterohepatic circulation, culminating in gallstone formation [58,63]. Hence, the disruption of the human flora equilibrium propels the progression of biliary tract cancer and its precancerous lesions.

## 4. Potential Role of Microbes in Chemotherapy and Immunotherapy for Biliary Tract Cancer

Immunotherapy stands as a pivotal approach for treating malignant tumors; its efficacy is influenced by intestinal flora and environmental factors [64,65]. The significance of the gut microbiome in various metabolic and signaling pathways, as well as its role in carcinogenesis, has been somewhat underestimated. Presently, it garners widespread attention as a critical avenue to bolster immunotherapy responses [66]. Demonstrated effects of the microbiota on cancer initiation, progression, and treatment response have hinted at potential contributions to susceptibility to specific cancers and possibly influencing treatment outcomes [67,68]. The correlation between the gut microbiome and the response to immune checkpoint inhibitors (ICI) is emerging as an intriguing area. The gut microbiome is linked to tumor immune resistance. The strategic combination of probiotics with ICI may aid in reshaping the microbiome [69]. 

Despite early clinical trials showing a relatively modest response rate of immunocheckpoint therapy (ICT) in cholangiocarcinoma [70], its therapeutic potential remains underexplored. Although potential MDSCs have been detected in cholangiocarcinoma, their exact role in its pathogenesis has remained unclear [52]. Recent studies have illuminated the potential of targeting MDSCs in other cancers to activate anti-tumor immune responses and amplify the effectiveness of ICTs [71,72]. Through the amalgamation of intestinal barrier dysfunction, the intestinal microbiome, and MDSC regulation, Zhang introduces a novel paradigm where inflammatory bowel diseases (IBD) and PSC may foster immunosuppression, thus molding the liver microenvironment conducive to cholangiocarcinoma progression [22]. The outcomes propose a variety of novel targets for intervening in cholangiocarcinoma growth, encompassing gram-negative intestinal bacteria, TLR4, CXCL1, CXCR2, and MDSCs themselves. This pivotal step in reversing the immunosuppressive microenvironment holds promise for heightening immunotherapy strategies, including ICT. In a study by Mao et al., the gut microbiome’s association with the clinical response to anti-PD-1 immunotherapy in hepatobiliary cancer patients was unveiled. Taxonomic signatures enriched in responders prove to be effective biomarkers for predicting clinical response and survival benefits from immunotherapy. This discovery offers a potential therapeutic target for modulating responses to cancer immunotherapy [73]. The microbiota may have exciting implications for therapeutic strategies for the microbiota-immune system axis in cholangiocarcinoma [74]. Microorganisms could potentially serve as drug targets for cholangiocarcinoma treatment, with fecal microbiota transplantation (FMT) potentially aiding in rectifying biological imbalances and optimizing anti-tumor immune responses. However, this therapeutic approach warrants further investigation.

## 5. The Role of Bacterial Metabolites in the Progression of Biliary Tract Cancer

In recent years, metabolomics has gained widespread utilization in hepatobiliary diseases, demonstrating significant advantages in understanding disease pathogenesis [75]. Analysis of the humoral metabolome is emerging as a promising diagnostic strategy, potentially linked to disease progression [76]. Bile acids are metabolized by enzymes produced by gut bacteria and are essential for maintaining a healthy gut microbiome and innate immunity. The liver–bile acid–microbiota axis plays an important role in gastrointestinal carcinogenesis [77,78]. Murakami et al. validated the connection between ICC and lipid metabolism as well as bile secretion, elucidating their participation in the metabolic reprogramming of ICC [79]. Furthermore, distinct alterations in plasma bile acid concentrations have been identified as potential diagnostic biomarkers for distinguishing cholangiocarcinoma from benign biliary diseases and healthy individuals [80]. Likewise, specific changes in serum metabolite levels contribute to the differentiation between ICC, HCC, and PSC [81]. 

Liu et al. observed high expression of S1PR2 in both rat and human cholangiocarcinoma cells, as well as in human cholangiocarcinoma tissues. They revealed that conjugated bile acids can promote the aggressive growth of cholangiocarcinoma cells through S1PR2 signaling [82]. In another study, Li et al. conducted GC-MS-based metabolomics experiments on ICC and intrahepatic bile duct stone (IBDS) pathological tissues, along with ICC para-carcinoma tissues. Their findings emphasized that the metabolic disparities between IBDS and ICC mainly revolve around linoleic acid metabolic pathways. Perturbations in the linoleic acid pathway might contribute to the potential malignant transformation of intrahepatic bile duct stones into ICC [83]. Chai et al. discovered that results from both in vitro and in vivo experiments strongly support the idea that *P. fungorum* demonstrates anti-tumor activity by modulating alanine, aspartate, and glutamate metabolism [23]. Figure 1 shows the relationship between human microbiome, metabolites and biliary tract cancer. Research on microbial metabolites in hepatobiliary diseases has gradually attracted attention, contributing to the understanding of disease mechanisms. 

## 6. Future Directions

The symbiotic microbial community within the human body represents a crucial element of human microbial equilibrium. Maintaining the stability of this microbial community might hold the key to averting tumorigenesis.

The current research on the microbiota in cholangiocarcinoma is constrained by factors such as sample quality and environmental interference. There can be significant biological variations among cholangiocarcinoma patients, including factors like tumor location, size, and differentiation level, leading to considerable variability in microbiota composition. Additionally, microbial communities are influenced by environmental factors such as diet and lifestyle, which may impact research outcomes. Microbiome profiling techniques may face limitations in terms of technical sensitivity and specificity, potentially resulting in the under-detection or misidentification of certain microorganisms. While alterations in microbiota composition associated with cholangiocarcinoma have been observed, further empirical evidence is needed to determine whether these microorganisms play a causal role or are merely correlated with the disease.

The intestinal flora, potentially serving as a non-invasive diagnostic biomarker for cholangiocarcinoma, holds promise as a tool for early diagnosis, prediction, and even as a future therapeutic target in biliary tract cancer. This approach could enhance the prospects of successful treatment. Envisioned is the continuous enhancement of cancer chemotherapy and immunotherapy efficacy through the utilization of gut flora. Distinct shifts in gut flora composition might lead to the production of specific metabolites that could be identified and exploited for early diagnosis. However, comprehensive comprehension of the mechanisms behind microbial-driven carcinogenesis remains a priority. More clinical and fundamental investigations are imperative. Moving forward, large-scale cohort studies are necessary to deeply analyze the microbiome using a multi-omics approach. Simultaneously, heightened focus should be directed towards elucidating the functions of the microbiome and its metabolites to gain a deeper understanding of the mechanisms underlying microbiome-related carcinogenesis. This understanding can then be harnessed to refine strategies for preventing, diagnosing, and treating biliary tract cancer. As our understanding of the intricate connection between the microbiome and biliary tract cancer deepens, the microbiome is poised to become a pivotal factor in cancer prevention and treatment. However, further research is indispensable to fully grasping its role and translating this knowledge into effective clinical strategies.

## Figures and Tables

**Figure 1 microorganisms-11-02598-f001:**
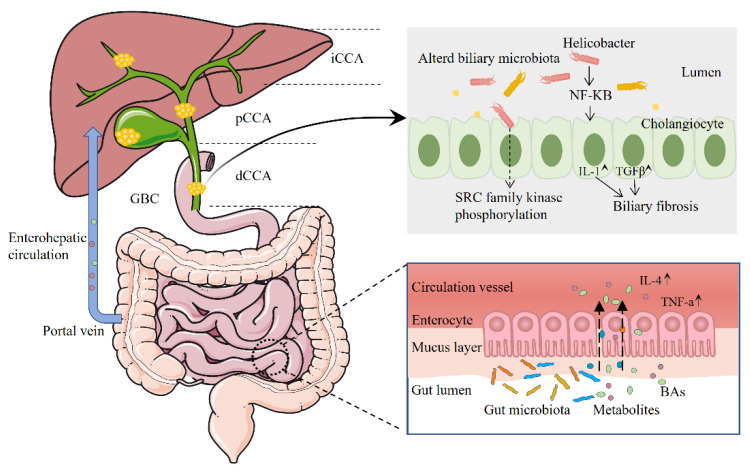
Association between the human microbiome, metabolites and biliary tract cancer.

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
