# Peer review of "Interplay between the Human Microbiome and Biliary Tract Cancer: Implications for Pathogenesis and Therapy"

_microorganisms, 2023, doi:10.3390/microorganisms11102598_

Round 1
Reviewer 1 Report
Dear authors,
The review with the title "Interplay Between the Human Microbiome and Biliary Tract Cancer: Implications for Pathogenesis and Therapy" sounded promising even from the title.
It is true that biliary tract cancers involves a multitude of risk factors and could potentially be influenced by microbial exposure, lately the microbiome due to the widespread application of next-generation sequencing technology has been hardly studied.
The review focuses upon review, the relationship between the digestive tract microbiome and biliary tract tumors, the role and significance of the digestive tract microbiome in the realm of biliary tract cancer treatment.
In Introduction section some representative bacteria from microbiome are introduced: Clostridia, Faecalibacterium, Coprococcus, Enterobacteriaceae, maybe more species should be discussed and more information about gut microbiome should be given.
The review extends its discussion with the correlation between microbiome and biliary tract cancer: intrahepatic cholangiocarcinoma, extrahepatic cholangiocarcinoma, gallbladder cancer in the first part corelating a lot of species with the modification.
Table 1 perfectly summarises the information, I think the authors should specify the Table in the text.
The topic continues with the effect of dysbiosis on biliary tract cancer and its precancerous lesions, the potential role of microbes in chemotherapy and immunotherapy for biliary tract cancer and the role of bacterial metabolites in the progression of biliary tract cancer.
The review ends very well with future directions.
I find the subject covered completely and I liked the review, the study design is well conducted and the references are well chosen.
I believe that to be easier to follow by anyone, the authors should make a table with all the acronims that are present into the text. I also believe that a table like table 1 with all the remaining references is appropriate.
I recommend publication after minor adjustments.
Author Response
Comments 1: [In Introduction section some representative bacteria from microbiome are introduced: Clostridia, Faecalibacterium, Coprococcus, Enterobacteriaceae, maybe more species should be discussed and more information about gut microbiome should be given.]
Response 1: Thank you for pointing this out. I mentioned the gut microbiome briefly in the introduction, and more information about the gut microbiome is described in detail both in the text and in the table.
Comments 2: Table 1 perfectly summarises the information, I think the authors should specify the Table in the text.
Response 2: Thank you for pointing this out. I have specified the table in the text on line 162.
Comments 3: I believe that to be easier to follow by anyone, the authors should make a table with all the acronims that are present into the text. I also believe that a table like table 1 with all the remaining references is appropriate.
Response 3: Thank you for pointing this out. I have listed all the acronims that appear in the text in lines 359 to 369.

Reviewer 2 Report
This manuscript provides a comprehensive and well-structured exploration of the role of the microbiome in biliary tract cancer, including cholangiocarcinoma and gallbladder cancer. It encompasses various aspects, from the influence of dysbiosis on cancer development to the potential applications of microbiome research in cancer diagnosis and treatment.
The manuscript extensively reviews the existing literature, integrating findings from both human and animal studies. The inclusion of mechanistic insights into how dysbiosis influences cancer progression adds depth to the discussion.
1. Overall, the manuscript is well-organized and the writing is clear. However, in some sections, particularly when discussing specific bacteria and metabolites, the text can become dense and might be challenging for readers without a strong background in microbiology.
2. This manuscript makes a significant contribution to the field of cancer research, particularly in the context of biliary tract cancers. It synthesizes existing knowledge and identifies key areas where further research is needed. Pleas cite Cells. 2023 Jan 19;12(3):370
3. The manuscript could benefit from a more explicit discussion of the limitations of existing research in this field. Acknowledging the gaps in knowledge and potential sources of bias in microbiome studies would provide a more balanced perspective.
4. The manuscript's section on future directions is particularly commendable. It outlines important areas for future research, emphasizing the need for clinical studies and mechanistic investigations to bridge the gap between findings and clinical applications.
In summary, this manuscript offers a valuable and insightful examination of the microbiome's role in biliary tract cancer. Its rigorous scientific approach, thorough coverage of the topic, and clear direction for future research make it a valuable resource for researchers and clinicians interested in this field.
The quality of English used in the provided manuscript is generally strong.
Author Response
Comments 2: This manuscript makes a significant contribution to the field of cancer research, particularly in the context of biliary tract cancers. It synthesizes existing knowledge and identifies key areas where further research is needed. Pleas cite Cells. 2023 Jan 19;12(3):370
Response 2: Thank you for pointing this out. I have cited Cells. 2023 Jan 19;12(3):370 in line 279 of the text.
Comments 3: The manuscript could benefit from a more explicit discussion of the limitations of existing research in this field. Acknowledging the gaps in knowledge and potential sources of bias in microbiome studies would provide a more balanced perspective.
Response 3: Thank you for pointing this out. I agree with this comment. In response to this issue, I further discuss the limitations of cholangiocarcinoma microbiological research in the section on future directions, see lines 317 to 327 of the text.
